# Quantitative Detection of *Plasmodium falciparum* Using, LUNA-FL, A Fluorescent Cell Counter

**DOI:** 10.3390/microorganisms8091356

**Published:** 2020-09-04

**Authors:** Muneaki Hashimoto, Kazumichi Yokota, Kazuaki Kajimoto, Musashi Matsumoto, Atsuro Tatsumi, Kenichi Yamamoto, Tomonori Hyodo, Kiichiro Matsushita, Noboru Minakawa, Toshihiro Mita, Hiroaki Oka, Masatoshi Kataoka

**Affiliations:** 1Health Research Institute, National Institute of Advanced Industrial Science and Technology (AIST), 2217-14, Hayashi-cho, Takamatsu, Kagawa 761-0301, Japan; kazumichi-yokota@aist.go.jp (K.Y.); k-kajimoto@aist.go.jp (K.K.); m-kataoka@aist.go.jp (M.K.); 2Konica Minolta, 1 Sakura-mashi, Hino, Tokyo 191-8511, Japan; musashi.matsumoto@konicaminolta.com (M.M.); atsuro.tatsumi@konicaminolta.com (A.T.); hiroaki.oka@konicaminolta.com (H.O.); 3Nitto Denko Corporation, 18, Hirayama, Nakahara-cho, Toyohashi, Aichi 441-3194, Japan; kenichi.yamamoto@nitto.com (K.Y.); tomonori.hyodo@nitto.com (T.H.); kiichiro.matsushita@nitto.com (K.M.); 4Institute of Tropical Medicine, Nagasaki University, 1-12-4 Sakamoto, Nagasaki 852-8523, Japan; minakawa@nagasaki-u.ac.jp; 5Department of Tropical Medicine and Parasitology, Juntendo University School of Medicine, 2-1-1 Hongo, Bunkyo-ku, Tokyo 113-8421, Japan; tmita@juntendo.ac.jp

**Keywords:** malaria, diagnosis, fluorescent cell counter, parasitemia

## Abstract

The microscopic examination of Giemsa-stained thin and/or thick blood films (Giemsa microscopy) is the standard method of malaria diagnosis. However, the results of the diagnosis significantly depend on the skills of clinical technicians. Furthermore, sample preparation and analysis are laborious and time-consuming. Therefore, in this study, we investigated if a commercially available fluorescent cell counter, LUNA-FL, was useful for the detection of *Plasmodium* parasite and the estimation of parasitemia. Whole blood samples from uninfected persons, spiked with *P. falciparum*-infected erythrocytes, were analysed. Most of the leucocytes and platelets were removed from whole blood samples with SiO_2_-nanofiber filters set on spin columns. The filtered samples were stained with acridine orange, and automatic detection, as well as counting of erythrocytes and parasites, were performed using LUNA-FL. Whole blood, with various levels of parasites, was analysed by Giemsa microscopy or with LUNA-FL to estimate parasitemia, and a comparative analysis was performed. The coefficient determination value of the regression line was high (*R^2^* = 0.98), indicating that accurate quantitative parasite detection could be performed using LUNA-FL. LUNA-FL has a low running cost; it is compact, fast, and easy to operate, and may therefore be useful for point-of-care testing in the endemic areas.

## 1. Introduction

The World Health Organization (WHO) recommends that all suspected malaria cases should have a parasite-based diagnosis, by either a microscopic analysis with Giemsa-stained blood films (Giemsa microscopy) or a rapid diagnosis test (RDT), prior to treatment with artemisinin-based combination therapy (ACT) [1]. For point-of-care testing (POCT), the gold standard or reference for malaria diagnosis is Giemsa microscopy. Giemsa microscopy with thick films and thin smears provides sensitivity and allows quantitation, respectively [2]. Accurate diagnosis with Giemsa microscopy is time-consuming and laborious, and it requires skilled microscopists [3]. RDTs are often used as the first-line investigation. In Africa, the most endemic area, a *Plasmodium falciparum*-only test based on the highly expressed histidine-rich protein 2 (HRP2) antigen is often used [2]. Diagnosis with RDTs is qualitative with high false-positive rates. RDTs can remain positive for several weeks after clearance because of persisting pitted (once-infected) erythrocytes [4]. It is necessary to develop a malaria diagnosis device, that is quantitative, easy to operate, and not time-consuming, for POCT.

Acridine orange (AO) staining has been optimised to reduce the time required to stain blood films [5,6,7,8,9]. Devices for automatic analysis of thick or thin Giemsa-stained blood films have also been developed [10]. They can be operated with smartphones, and may be useful for POCT; however, skilled technicians are required for the preparation of blood films. Furthermore, the preparation of thick blood films involves more than 1 h in an incubator at 37 to 40 °C, or overnight at room temperature.

In order to perform highly sensitive and quantitative malaria diagnosis, some devices, that can detect fluorescent nuclear-stained parasites in infected erythrocytes by forming a monolayer on microchambers or a microplate [11,12,13,14], or by using flow cytometers [15], have been developed. These devices have a microscope with a digital camera or a charge-coupled device camera for the detection of erythrocytes and/or labelled parasites, and hence, they may be unsuitable for POCT except in hospitals located in urban areas. The limitations of these devices include: (1) Unstable power supply in endemic areas, (2) High cost which makes it unaffordable for people in developing countries, and (3) Fragility which makes maintenance difficult.

Recently, we developed a novel diagnosis device by applying a Blu-ray optical system [16]. The device consists of a disposable compact disk (CD)-like cassettes and a CD player-like image reader. Approximately one million erythrocytes form a monolayer on the cassette, and nuclear-stained parasites are detected with an image reader followed by the automatic determination of the infection rate (parasitemia). However, the device has some limitations. Firstly, although nine samples can be analysed at the same time, parasite counting takes a moderately long time (approximately 40 min). Secondly, computers that are able to process a large amount of image data are required.

In this study, we attempted to develop a novel malaria diagnosis device that is more suitable for POCT in endemic countries. Whole blood of healthy donors spiked with in vitro cultured parasite-infected erythrocytes was prepared, and parasitemia was estimated using a fluorescent cell counter, LUNA-FL (Logos Biosystems, Inc., Gyeonggi-do, Korea). LUNA-FL is a compact and computer-integrated device, and rapid cell counting can be performed by AO staining. Actually, rapid, easy-to-operate, and accurate quantitative malaria detection was possible using LUNA-FL, suggesting that malaria diagnosis can be performed in endemic countries using the device.

## 2. Materials and Methods

### 2.1. Parasite Culture

*P. falciparum* (3D7 strain) was cultured as previously described [16]. We synchronised the parasites at the ring stage using 5% D-sorbitol and harvested highly synchronous ring-stage parasite-infected erythrocytes (>95%) [17]. Parasitemia was estimated by microscopic analysis with Giemsa-stained thin blood films. More than 5000 erythrocytes were counted for the estimation of parasitemia [% parasitemia = (parasitised RBCs/total RBCs) × 100].

### 2.2. Preparation of Spin Columns

Empty spin columns were purchased from Nippon Gene Co., Ltd. (Tokyo, Japan). SiO_2_-nanofiber (NF) filters (250 µm thick) (Nitto Denko Corp., Osaka, Japan) were punched at 7 mm diameters. Two sheets of the punched filters were set on the spin column and fixed with an O-ring.

### 2.3. Erythrocytes, Leucocytes, and Platelet Counting

Fresh whole blood samples drawn by finger pricking from Japanese healthy volunteers who have never been infected with malaria parasites were collected in BD Microcontainer blood collection tubes (K2EDTA, Lavender, Becton, Dickinson Co., Franklin Lakes, NJ, USA). The whole blood sample (200 µL) was diluted 25-fold with phosphate buffered salts (PBS) and applied to the spin column followed by centrifugation at 400 g for 30 sec. To count the erythrocytes, the filtered blood sample was diluted 100-fold with PBS, and 10 µL of the aliquot was applied to a photon slide (Logos Biosystems, Inc., Gyeonggi-do, Korea). The number of erythrocytes was counted with LUNA-FL using the “Bright-Field Counting mode” as described in the manual. The leucocytes in the filtered blood samples were stained with AO/PI Cell Viability Kit (Logos Biosystems, Inc.) in order to count the leucocytes and 10 µL of the aliquot was applied to a photon slide. The number of leucocytes was counted with LUNA-FL using the “Fluorescence Cell Counting mode”. The exposure times for the green fluorescence and red fluorescence were set to Levels 5 and 1, respectively. For the specific detection of leucocytes using LUNA-FL at the exposure times, the threshold for both the green fluorescence and the red fluorescence in the protocol menu was set to Level 5, and the size gating was from 5 to 30 µm. Counting of the platelets was performed using a flow cytometer (BD FACS Calibur, Becton, Dickinson Co.) with a fluorescein isothiocyanate (FITC)-labelled antihuman CD61 monoclonal antibody (Beckman Coulter, Inc., Brea, CA, USA). The percentage of platelet in the blood cells was estimated by counting 100,000 blood cells.

### 2.4. Detection of Parasite-Infected Erythrocytes Using LUNA-FL

To analyse parasite-infected erythrocytes using LUNA-FL, an appropriate volume of purified parasite-infected erythrocyte suspension was added to whole blood to obtain 0.4, 0.2, 0.08, 0.04, and 0.02% parasitemia at 2% haematocrit. Parasitemia was estimated by thin-smear microscopy; each sample (200 µL) was filtered with the spin column. To estimate parasitemia with LUNA-FL, the filtered blood sample was diluted 100-fold with PBS and the number of erythrocytes was counted using the “Bright-Field Counting” mode. To count the parasites, 9 µL of the filtered sample was stained with 1 µL of AO/PI Cell Viability Kit and placed on a photon slide. The number of parasites was counted using the “Fluorescence Cell Counting” mode. The exposure times for green fluorescence and red fluorescence were set to Level 12 and Level 1, respectively. The thresholds of the green fluorescence and the red fluorescence in the protocol menu was set to Level 8 and Level 5, respectively, and the size gating was 1 to 9 µm. A high threshold setting does not mark the signals with low fluorescent intensity as those of the parasite nuclei.

### 2.5. Statistical Analysis

Statistical analysis was performed with SigmaPlot Version 13 (Systat Software, Inc., San Jose, CA, USA).

### 2.6. Ethics Statement

Ethics approvals were obtained from the National Institute of Advanced Industrial Science and Technology ethics committee (No. 2018-0204) on 4 December, 2019.

## 3. Results

### 3.1. Protocol for Detection of Parasite Using LUNA-FL

Fluorescent nuclear-stained signals for *P. falciparum* ring forms are significantly smaller and weaker than that for leucocytes [11,16]. The photon slides were used as disposable slides for cell counting for LUNA-FL (Figure 1A). They were developed for highly sensitive detection of fluorescent signals. The photon slide has two chambers, and 10 µL of a sample can be applied to each chamber. Automatic cell counting with bright-field images can be performed using LUNA-FL. The erythrocytes were counted in the bright-field images, as they were not stained with the nuclear stain (Figure 1Ba). Too many erythrocytes in an image resulted in inaccurate cell counting. Some erythrocytes aggregated and/or overlapped, and hence, the 25-fold diluted blood samples at approximately 1% haematocrit were further diluted 100-fold with PBS. The diluted sample was applied to Chamber A, and the number of erythrocytes was counted. It was important to adjust focus using the focus knob on LUNA-FL before counting the samples, for accurate counting of erythrocytes and the parasites.

Blood samples were 25-fold diluted with PBS, and 9 µL of the diluted sample was stained with 1 µL of AO solution as described in “Materials and Methods” section. The blood sample (10 µL) was applied to Chamber B (Figure 1A), and the chamber was set to the “Fluorescent Image Analysis” mode (Figure 1Bb). Although the default exposure time for the counting of leucocytes is Level 5, the exposure time for parasite detection was set to Level 12 for green fluorescence. The exposure time for red fluorescence was set to Level 1 so as to not detect any fluorescent signals. After counting the parasites, the slide was moved slightly (note the arrow in Figure 1Bb) to change the observation visual field (Figure 1Bc), and the counting of the parasites was also performed in the field. In the counting protocol, the counting of erythrocytes and parasites in two visual fields followed by the estimation of parasitemia was possible.

### 3.2. Development of Spin Columns for the Purification of Erythrocytes

Malaria diagnosis can be performed using devices that can detect unicellular parasites in erythrocytes, and the purification of erythrocytes from whole blood is crucial for such devices. Especially, the contamination of samples by leucocytes and platelets which have nucleic acids could increase the background signals. It has been reported that SiO_2_-NF filters are useful for the removal of leucocytes [11]. Two sheets of SiO_2_-NF filters (250 µm thickness) were set on an empty spin column and fixed with an O-ring (Figure 2).

The diluted blood sample (25-fold, 200 µL) was applied to the spin column. After centrifugation at 400× *g* for 30 s, the filtered sample resulted in a purified erythrocyte sample. Using LUNA-FL, the number of erythrocytes counted before filtration was 3.8 ± 0.28 × 10^6^ erythrocytes/µL, and that after filtration, in filtrate collected in the collection tubes, was 1.6 ± 0.14 × 10^6^ erythrocytes/µL (Figure 3A). Upon filtration, 58.6% of erythrocytes were removed on average. In addition, the number of leucocytes before filtration was 7.4 ± 0.52 × 10^3^ cells/µL, and no leucocytes were detected in the filtrate in the collection tubes after filtration in this assay method (Figure 3B). The number of platelets in the samples was also investigated using a flow cytometer with an anti-CD61 antibody (Figure 3C). The number of platelets before filtration was 2.8 ± 0.13 × 10^5^ platelets/µL, and that after filtration was 0.007 ± 0.0027 × 10^5^ platelets/µL. After filtration, almost all the leucocytes were removed, and 97.8% of platelets were removed on average.

The effect of filtration on parasitemia was also investigated (Figure 3D). In vitro parasite culture at 2% haematocrit was prepared. Parasitemia before and after filtration was 4.54 ± 0.32% and 4.50 ± 0.50%, respectively, indicating that filtration essentially had no effect on parasitemia (*p* = 0.55). Parasite-infected erythrocytes and uninfected erythrocytes may bind to SiO_2_-NF filters in the same manner. Taken together, it was possible to remove leucocytes and platelets from whole blood effectively and rapidly, without influencing parasitemia, by using the spin column with SiO_2_-NF filters.

### 3.3. Detection of Parasites Using LUNA-FL

Typical images of whole blood sample diluted 50-fold with PBS and whole blood sample diluted 25-fold with PBS after filtration are shown in Figure 4A and Figure 4B, respectively. Figure 4C shows typical images of whole blood spiked with parasite culture diluted 50-fold with PBS (parasitemia was adjusted to 0.06 %) and Figure 4D shows typical images of whole blood spiked with parasite culture diluted 25-fold with PBS after filtration. The bright-field images (upper panels) and the fluorescent images stained with AO (lower panels) of each sample are shown.

Bright-field images indicated that erythrocytes form a monolayer with relatively high density. Although many leucocytes were detected in the whole blood sample (Figure 4A), all the cells were removed after filtration (Figure 4B). In whole blood spiked with parasite-infected erythrocytes sample (Figure 4C), the parasite nuclei stained with AO were detected as small signals (indicated by arrows in Figure 4C); however, this was not detected in the whole blood samples (Figure 4A). After the filtration of whole blood spiked with the parasites, leucocytes were removed, but the parasites were not removed (Figure 4C,D).

Considering that only ring forms of *P. falciparum* usually appear in peripheral blood, the in vitro cultured parasites were tightly synchronised and all were ring forms. The fluorescent signals corresponding to the ring forms were significantly smaller than that of leucocytes, and the fluorescent intensity of the parasites was also significantly weaker than that of leucocytes (Figure 4E).

### 3.4. Quantitativity and Limit of Detection (LOD) of LUNA-FL for Parasitemia Calculation

To investigate the quantitativity and the LOD of LUNA-FL for the estimation of parasitemia, whole blood samples spiked with parasite culture having different percentages of parasitemia were prepared. Final parasitemia of each sample was 0.4, 02, 0.08, 0.04, 0.02, and 0%. Each sample was filtered with the SiO_2_-NF spin column, followed by staining with AO and calculation of parasitemia using LUNA-FL as described in Figure 1 and ‘Materials and Methods’ section. The number of erythrocytes counted in one assay was 61,690 ± 8,280.

Each of the samples with 0% parasitemia was analysed 10 times, and each of the other samples was analysed five times. Parasitemia was estimated by Giemsa microscopy and LUNA-FL, and both were compared (Figure 5). The coefficient of determination value (*R^2^*) was 0.98, indicating that parasitemia was accurately calculated using LUNA-FL. The background by LUNA-FL was 0.0009 ± 0.00012%. LOD was determined as 3SD/S (SD is the standard deviation of the background, and S is the slope value of the approximation straight line), and was found to be 0.0043%.

## 4. Discussion

In this study, a commercially available fluorescent cell counter, LUNA-FL, was used for the determination of parasitemia. Blood samples with appropriate parasitemia were prepared, and whole blood from uninfected volunteers spiked with the parasites cultured in vitro was analysed for the evaluation of the device. By using spin columns with SiO_2_-NF filters that could almost completely remove leucocytes and platelets from whole blood samples, purified erythrocyte samples were prepared quickly. The samples were stained with AO, and we found that accurate parasite detection could be performed easily using LUNA-FL.

The estimation of parasitemia is crucial for malaria diagnosis. As the severity of malaria and/or the effect of antimalarial drug treatment should be monitored, the calculation of parasitemia is very important for malaria diagnosis. Microscopic analysis with thin or thick blood films remains the gold standard diagnosis method. However, to perform Giemsa microscopy, fully trained and skilled technicians must be employed, and the diagnosis is largely skill- and equipment- (e.g., microscope) dependent [10]. For Giemsa microscopy, a light microscope with a 100× objective lens and highly paid microscopists are required. Thus, not only the front-end cost, but also the running cost is very high. Considering the cost of maintaining skilled microscopists in medical institutions, the running cost of diagnosis using LUNA-FL may be much lower than that using Giemsa microscopy. The number of erythrocytes that can be counted on thin blood films in a microscope field is few, and hence, the estimation of parasitemia with Giemsa microscopy is time-consuming. Similarly, the preparation of thick blood films is also time-consuming because the erythrocytes on the films have to be haemolysed completely and the films have to be fully dried before analysis. By using LUNA-FL, these issues may be addressed as follows: (1) Blood samples could be applied to photon slides quite easily, and skills for the preparation of blood films are not required. (2) Calculation of parasitemia by automatic erythrocyte and parasite counting using LUNA-FL is easily and quickly performed. Even if technicians are not experts, stable and correct results can be obtained. (3) The main body of LUNA-FL is small (220 × 210 × 90 mm) and light (1.8 kg). Therefore, it can be put in small medical offices. (4) The average time taken for parasitemia calculation is approximately 5 min, which may enable the quick start of malaria treatment. Because LUNA-FL is not manufactured for developing countries, the machine must be plugged in constantly. An adequate power supply is not available in many medical institutions in malaria-endemic countries. In this case, a solar battery may be useful.

RDTs for HRP2 are often used for malarial diagnosis in Africa. However, diagnosis with RDTs is qualitative with high false-positive rates in the most endemic areas. Furthermore, *P. falciparum* parasites with a deletion of the *hrp2* gene yield false-negative RDTs [18]. As diagnosis using LUNA-FL detects the parasites themselves (i.e., not the antigen) and parasite nuclei (i.e., not the *hrp2* gene), LUNA-FL might be more suitable for malaria diagnosis in endemic areas.

The comparative analysis between LUNA-FL and Giemsa microscopy in Figure 5 revealed that the equation of the approximation straight line was y = 0.86x + 0.0042. The equation slope value was close to 1, indicating that the underestimation of counted parasites did not occur in the analysis with LUNA-FL. In a previous study, the LOD of Giemsa microscopy with thin blood films by skilled technicians was ~0.004% (~200 parasites/µL of blood) [19]. The LOD of the analysis with LUNA-FL using the protocol in this study was 0.0043% (215 parasites/µL of blood), and the LOD was equivalent to that of the gold standard method. Two image fields were analysed with a photon slide, and it may be possible to increase the number of analysed erythrocytes by increasing the number of photon slides analysed, but the cost of diagnosis will also increase. Therefore, a device that is able to capture many image fields on a slide should be developed. In addition, it may be necessary to modify the main body of LUNA-FL and/or the slides.

The findings of this study showed that the removal of leucocytes and parasites from whole blood samples could certainly be performed by using spin columns with two sheets of SiO_2_-NF filters fixed with O-ring. The fixation of the filters with the O-ring was performed easily and quickly; however, a centrifuge was also required. For malaria diagnosis in facilities that are not equipped, the development of devices that can filter samples without a centrifuge is required. Previously, a push-column-type device that can filter without centrifugation was developed [11]. The setting of the SiO_2_-NF filter to the push-column was performed with glue that hardens upon UV exposure. In our previous study, we observed that the preparation of the spin columns was time-consuming and laborious, and there was an evident difference in the leucocyte and/or platelet removal efficiency of the columns and a possibility that the dregs of the glue contaminated the blood samples (unpublished data). Taken together, the development of a push-column with SiO_2_-NF filters fixed with an O-ring may be important.

Two kinds of fluorescent signals (green and red) can be detected using LUNA-FL to distinguish between live and dead cells. The counting of dead cells is not required in malaria diagnosis, and hence, the exposure time for red fluorescent signals was set not to detect any cell and parasite. If LUNA-FL can be modified to develop a device that can detect only a green fluorescent signal, the cost or the time required for malaria diagnosis should be significantly reduced.

The cost of one assay was approximately two dollars. Because the photon slides are manufactured from acryl, initial costs may be very low. The slides are mainly manufactured for and sold to researchers in developed countries. We expect the selling price could be further decreased if the slides were manufactured in Africa, which may result in decreased labour and transportation costs. Furthermore, because it took only approximately 1 min to capture and analyse a field image, and three field images were required to be analysed in this protocol, it took approximately 5 min for the measurement of parasitemia. During that period of time, we were able to analyse an average of 61,690 erythrocytes, indicating that malaria diagnosis using LUNA-FL may be useful for POCT in endemic areas.

## Figures and Tables

**Figure 1 microorganisms-08-01356-f001:**
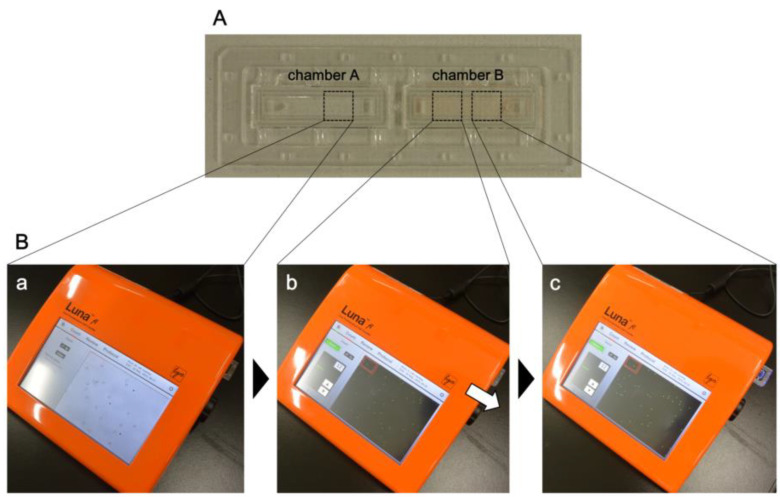
Methods for the estimation of parasitemia using LUNA-FL. (**A**) Photon slides were used for the detection of parasites using LUNA-FL. Blood samples diluted for erythrocyte counting were applied to Chamber A. Blood samples diluted and acridine orange (AO)-stained for parasite counting were applied to Chamber B. (**B**) First, bright-field images in Chamber A were captured, to adjust focus, followed by the counting of erythrocytes (**a**). Second, two fields of fluorescent images in Chamber B were captured to count the parasites (**b**,**c**). Parasitemia was estimated from the data obtained from Chamber A and Chamber B.

**Figure 2 microorganisms-08-01356-f002:**
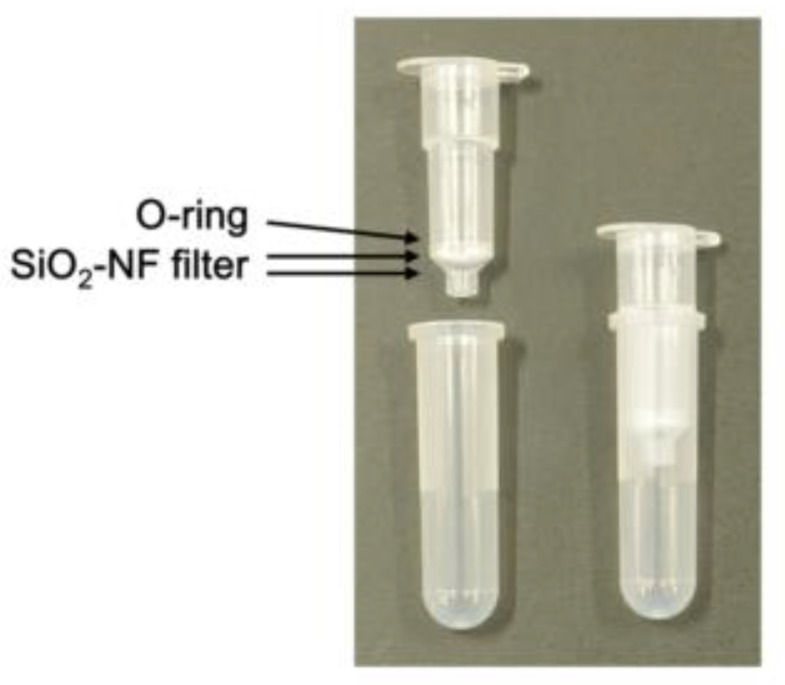
Spin columns with SiO_2_-NF filters for the removal of leucocytes and platelets from whole blood. Two sheets of SiO_2_-NF filters were set on an empty column and fixed with an O-ring (left). The column was set on a collection tube (right).

**Figure 3 microorganisms-08-01356-f003:**
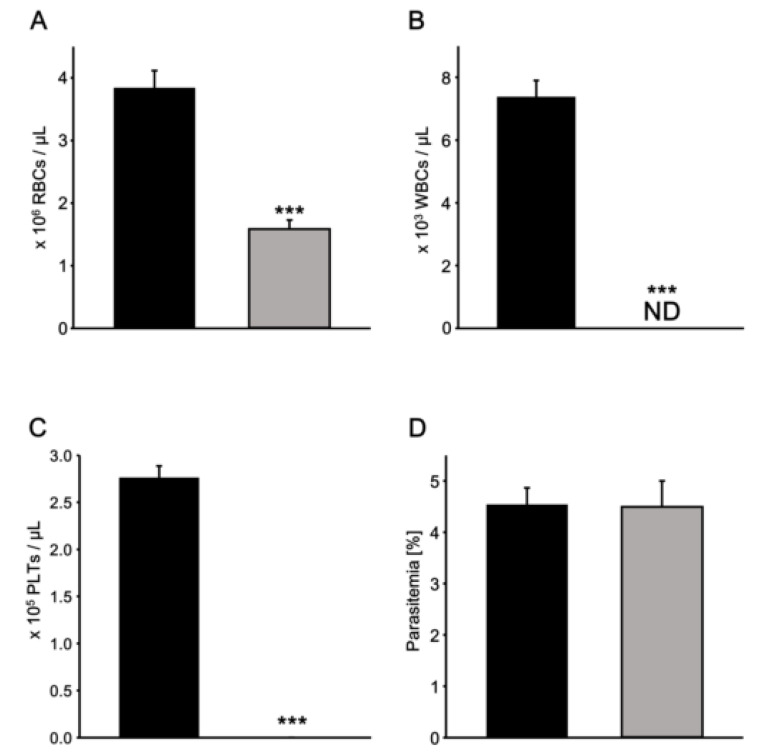
Evaluation of spin columns for the filtration of whole blood and parasite culture. The number of erythrocytes (**A**), leucocytes (**B**), or platelets (**C**) was analysed before (black bars) or after (grey bars) filtration of whole blood. Parasitemia of in vitro culture was determined before or after filtration (**D**). *** *p* < 0.001 (Student’s *t*-test, *n* = 5).

**Figure 4 microorganisms-08-01356-f004:**
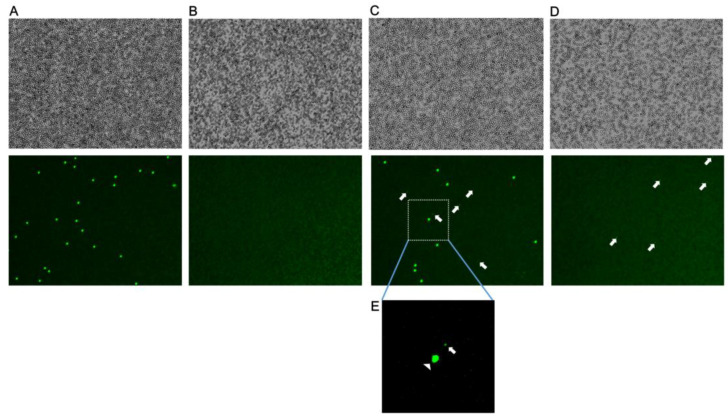
Bright-field and fluorescent images captured using LUNA-FL. Whole blood (**A**), whole blood after filtration (**B**), whole blood spiked with parasite-infected erythrocytes (**C**), or whole blood spiked with parasite-infected erythrocytes after filtration (**D**) were analysed using LUNA-FL. Each sample was stained with AO. Typical bright-field images (upper panels) and the fluorescent images (lower panels) are shown. The signals corresponding to the parasites are indicated by arrows. (**E**) An enlarged image of (C) is shown. A leucocyte is indicated by an arrowhead, and a parasite is indicated by an arrow.

**Figure 5 microorganisms-08-01356-f005:**
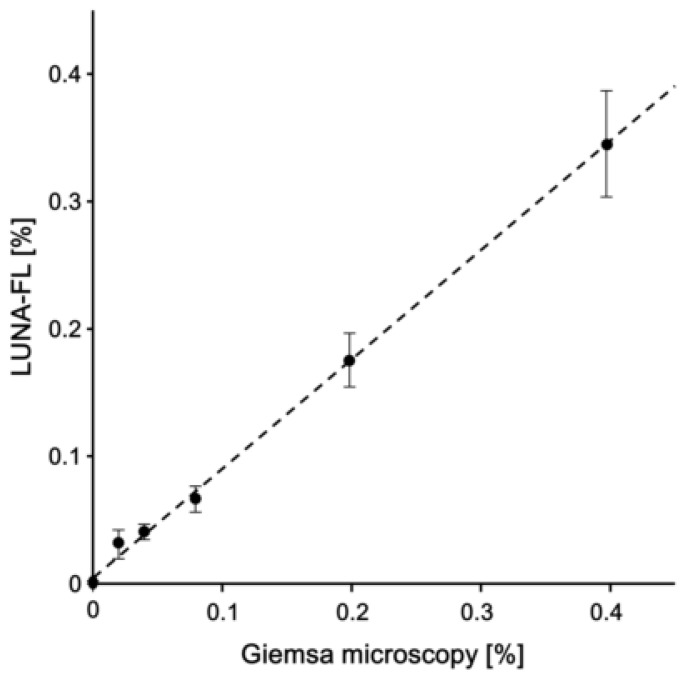
Comparative analysis of the parasitemia estimated with LUNA-FL and Giemsa microscopy. Linear regression analysis was used. Data are expressed as the mean ± SD for five different experiments.

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
