# Peer review of "Quantitative Detection of Plasmodium falciparum Using, LUNA-FL, A Fluorescent Cell Counter"

_microorganisms, 2020, doi:10.3390/microorganisms8091356_

Round 1

Reviewer 1 Report

This is a report investigating the use of a commercially available fluorescent counter, the LUNA-FL, to quantify malaria parasitemia. The group has previously published on malaria diagnostics and continues to explore possible alternatives to current malaria diagnostic modalities. The report is interesting and novel. The work is rigorously performed and described well.

The major barrier to more universal acceptance of a diagnostic technique such as this is the ease, rapidity and relatively low cost of RDTs. The authors, therefore, need to address the benefits of this technique in more detail as compared to RDTs. Is there a special ‘niche’ for this technique where RDTs fail? Could it be used as a secondary screen in certain settings where the high levels of false positives seen with RDTs are unacceptable? They could also mention the fact that this technique is not dependent on the presence of HRP2. There has been a lot of discussions in the malaria field regarding the emergence of HRP2 negative parasites and the consequences of that on malaria diagnosis.

Coupled with defining a niche for the machine, the authors should be encouraged to discuss the cost compared to RDTs. The current cost is prohibitive compared to RDTs. They should speculate as to whether the cost of the photon slides could be decreased. Are two measurements at fluorescent mode always necessary? How much accuracy would be lost if only one were performed? And how much would that lower the cost?

In addition, the electricity requirements of the machine need to be discussed. As discussed for the SiO2 filter, it would be ideal to have an electricity independent system. Barring that, however, there could be the possibility that the machine was to be charged over night and then taken to a health center that is lacking electricity. Does the machine hold a charge? Or must it be plugged in constantly? Could it run off a battery pack? And for how long? or for how many tests? These would be interesting additions to the Discussion section to perhaps make use of the technique in a low resource setting more feasible.

MINOR:

Line 19 “Correspondence” repeated

Line 172 – should be “on average”

Reviewer 2 Report

The article " Quantitative detection of Plasmodium Falciparun using LUNA FL, a fluorescent cell counter" is good a attempt in malaria field done by Hashimoto et.al., but I think author could improve this manuscript.

Here is some of my question and suggestion:-

Author should explain the level 5, 1 in context of flourescence.

what software author used for counting the parasitemia? how the instrument count the parasite or the counting was done by eye?

I feel in result part of this manuscript, author explain all the protocol which might be goes into the material and method section.

Author should clear what part of the filtration they used for counting e.g. liquid part which get in column or author recover from membrane.

Author showed before and after filtration, 58.6% erythrocyte removed (not clear they loss it or got it),

If the differences between before and after filtration is about half then how you explain that parasite is same after and before filtration because parasite is in erythrocyte?

What synchronization method author used?

Round 2

Reviewer 2 Report

Author answers all the question which I had. but I still don't understand if you filter normal erythrocyte you loose about 58% (line 171) but when you filter parasite infected erythrocyte how you get about 100% infected erythrocyte? Can you explain it briefly?

Author Response

Because parasite-infected and uninfected erythrocytes may bind to SiO2-NF filters in the same manner (lines 181–182), we cannot get 100% of the infected erythrocytes after filtration. Therefore, a portion of infected erythrocytes present in blood cannot be recovered by filtration. However, for diagnosis of malaria, it is important that filtration does not alter parasitemia (line 180, Fig. 2D). Therefore, diagnosis of malaria in patients with parasitemia above the limit of detection (LOD, line 277) might be possible by using LUNA-FL.